# Human Bone Marrow Mesenchymal Stromal Cell-Derived CXCL12, IL-6 and GDF-15 and Their Capacity to Support IgG-Secreting Cells in Culture Are Divergently Affected by Doxorubicin

Gintare Lasaviciute [1], Anna Höbinger [2], Dorina Ujvari [3], Daniel Salamon [2], Aisha Yusuf [3,4], Mikael Sundin [5], Eva Sverremark-Ekström [1], Rayan Chikhi [6], Anna Nilsson [2] and Shanie Saghafian-Hedengren [2,*]

[1]   Department of Molecular Biosciences, The Wenner-Gren Institute, Stockholm University, 106 91 Stockholm, Sweden; gintare.lasaviciute@su.se (G.L.); eva.sverremark@su.se (E.S.-E.)
[2]   Division of Paediatric Oncology and Paediatric Surgery, Department of Women's and Children's Health, Karolinska Institutet, 171 77 Stockholm, Sweden; anna.hobinger@ki.se (A.H.); daniel.salamon@ki.se (D.S.); anna.nilsson.1@ki.se (A.N.)
[3]   Department of Women's and Children's Health, Karolinska Institutet, 171 77 Stockholm, Sweden; dorina.ujvari@ki.se (D.U.); ay327@cam.ac.uk (A.Y.)
[4]   Medical Research Council, Cancer Unit, University of Cambridge, Cambridge CB2 0XZ, UK
[5]   Division of Paediatrics, Department of Clinical Science, Intervention and Technology, Karolinska Institutet, 141 52 Huddinge, Sweden; mikael.sundin@ki.se
[6]   Department of Computational Biology, Institute Pasteur, 75015 Paris, France; rayan.chikhi@pasteur.fr
*   Correspondence: shanie.hedengren@ki.se; Tel.: +46-707-185-606

**Abstract:** Various subsets of bone marrow mesenchymal stromal cells (BM MSCs), including fibroblasts, endothelial, fat and reticular cells, are implicated in the regulation of the hematopoietic microenvironment and the survival of long-lived antibody-secreting cells (ASCs). Nowadays it is widely acknowledged that vaccine-induced protective antibody levels are diminished in adults and children that are treated for hematological cancers. A reason behind this could be damage to the BM MSC niche leading to a diminished pool of ASCs. To this end, we asked whether cell cytotoxic treatment alters the capacity of human BM MSCs to support the survival of ASCs. To investigate how chemotherapy affects soluble factors related to the ASC niche, we profiled a large number of cytokines and chemokines from *in vitro*-expanded MSCs from healthy donors or children who were undergoing therapy for acute lymphoblastic leukemia (ALL), following exposure to a widely used anthracycline called doxorubicin (Doxo). In addition, we asked if the observed changes in the measured soluble factors after Doxo exposure impacted the ability of the BM niche to support humoral immunity by co-culturing Doxo-exposed BM MSCs with *in vitro*-differentiated ASCs from healthy blood donors, and selective neutralization of cytokines. Our in vitro results imply that Doxo-induced alterations in BM MSC-derived interleukin 6 (IL-6), CXCL12 and growth and differentiation factor 15 (GDF-15) are not sufficient to disintegrate the support of IgG-producing ASCs by the BM MSC niche, and that serological memory loss may arise during later stages of ALL therapy.

**Keywords:** human bone marrow mesenchymal stromal cells; anthracycline; plasma cell niche; cytokines; chemokines; antibody-secreting cells

## 1. Introduction

Bone marrow (BM) mesenchymal stem cells are a rare population of multipotent non-hematopoietic progenitor cells which in vivo and in vitro possess the ability to differentiate towards lineages of mesenchymal tissues such as bone, fat and cartilage, and the constitutes of hematopoietic microenvironment following transplantation [1–6]. Various subsets of BM mesenchymal stromal cells (MSCs) including fibroblasts, endothelial, fat and reticular cells are implicated in the regulation of the hematopoietic microenvironment. This includes

early B-cell lineage progression as well as the sustenance of antibody-producing plasma cells (PCs) by provision of MSC-derived growth factors, cytokines and chemokines as well as delivery of signals through cell receptor interactions [7–11].

PCs are divided into short-lived or long-lived (LL) on the basis of their longevity. As a rule, short-lived PCs or plasmablasts divide and have a lifespan of approximately one week, while LLPCs are terminally differentiated non-dividing cells surviving for months to years in the BM. Furthermore, it is not entirely clear if LLPCs originate from short-lived PCs, or if the two PC types are distinct cell populations [12], but the term antibody-secreting cells (ASCs) can be used to cover both short-lived PCs and LLPCs [12]. Two BM MSC-derived factors central for the maintenance of LLPCs in their BM niche are the cytokine interleukin 6 (IL-6) and the chemokine CXCL12. IL-6 interacts with the IL-6 receptor on PCs, and together with either a proliferation-inducing ligand (APRIL) or stromal cell-soluble factors, IL-6 has been shown to be mandatory for the in vitro generation and survival of human LLPCs [13]. CXCL12 mediates PC homing to the BM niche via interaction with the chemokine receptor CXCR4 expressed on LLPCs and plasmablasts [14]. In addition to its role as chemoattractant, CXCL12 acts as a docking molecule for eosinophils [15] and megakaryocytes [16] that constitute the accessory cell component of the PC niche through their delivery of IL-6 and APRIL [13]. Aside from their central role as regulators of normal hematopoietic cells, MSCs have been implicated in the pathogenesis of hematological malignancies such as myelodysplastic syndromes, acute leukemia and PC neoplasm multiple myeloma [17–19]. In myeloma patients, mesenchymal stem cell-derived growth and differentiation factor 15 (GDF-15) instigates the expansion and survival of malignant PCs [19,20], while less is known about the role of GDF-15 in the healthy PC niche.

BM MSCs have slow turnover in vivo and are therefore less sensitive to high-dose chemotherapy than hematopoietic cells. However, previous studies show that cytotoxic therapies and radiotherapy also target stromal cell function with long-term impairment of clonogenic stromal cell capacity (reviewed in [21]). There is still a knowledge gap on the extent of damage to the BM MSC niche in general, and impairments imposed by cancer therapy to the LLPC survival niche in particular [21]. Nowadays, it is widely acknowledged that vaccine-induced protective antibody levels are diminished in adults and children that are treated for hematological cancers [22–25]. A reason behind this could be damage to the BM MSC niche leading to a diminished pool of LLPCs, similar to what we have earlier observed in pediatric acute lymphoblastic leukemia (ALL) patients [22]. To this end, we asked whether cytotoxic cell treatment alters the capacity of human BM MSC to support the survival of ASCs in culture. To investigate how chemotherapy affects soluble factors related to the PC niche, we profiled a large number of cytokines and chemokines from *in vitro*-expanded MSCs from healthy donors or children who were undergoing ALL therapy, following exposure to a widely used anthracycline called doxorubicin (Doxo). In addition, we asked if the observed changes in the measured soluble factors after Doxo exposure impacted the ability of the PC niche to support humoral immunity by coculturing Doxo-exposed BM MSCs with *in vitro*-differentiated ASCs from healthy blood donors. Our results suggest that alterations in BM MSC-derived IL-6, CXCL12 and GDF-15 are not sufficient to disintegrate the support of IgG-ASCs by the PC niche *in vitro*.

## 2. Materials and Methods

### 2.1. Sampling and Ex Vivo Culture of Human BM Derived MSCs

MSCs were isolated from BM aspirates taken from the iliac crest of three ($n = 3$) healthy adult donors, aged 20–40 years, who served as healthy controls (HC), and from three ($n = 3$) children, aged two, three, and twelve, at day 106 of treatment against pre-B ALL with the NOPHO ALL-2000 protocol (from now on referred to as ALL). All donors in the ALL group were in complete response with minimal residual disease (MRD) <0.1%, evaluated by flow cytometry according to published work [26]. Additional ($n = 2$) BM MSCs derived from healthy adults were purchased from ATCC-LGC Standards (PCS-500-

012) to serve as HC. Isolation and in vitro culture of MSCs from BM aspirates occurred as published earlier [27]. Briefly, BM MSCs were cultured in human NH Expansion medium (Miltenyi Biotec, Bergisch Gladbach, Germany) for 10–16 days, with exclusion of non-adherent cells and replacement to fresh medium every 3 or 4 days. When cultures were nearly confluent, BM MSCs were detached by Trypsin-EDTA (Gibco, Invitrogen, Carlsbad, CA, USA), counted and re-plated at a density of $4 \times 10^3$ cells/cm$^2$ (passage 1). BM MSCs at passage 1–3 were frozen until further analysis. The Doxo treatment model was designed with respect to how well BM MSCs tolerated the drug in vitro in keeping with optimal viability and morphology and restricted passage numbers. For Doxo exposure, BM MSCs were thawed and expanded in cell culture flasks (Corning Inc., Corning, NY, USA). Thereafter $0.1 \times 10^6$ cells with 2 mL culture medium were transferred to six-well culture plates and kept until cell confluence reached 80% (after two days, approximately). After removal of medium, BM MSCs were incubated for 2 h with Doxo (Sigma-Aldrich, St. Louis, MO, USA) at a concentration of 0.5 or 1.0 µg/mL. BM MSCs in cell culture medium alone served as non-exposed controls. Next, BM MSCs were washed with PBS to remove Doxo, followed by addition of fresh medium. After overnight rest, the same cycle of Doxo treatment, removal, wash and medium refill was repeated, but this time with 3 h Doxo exposure. BM MSCs were thereafter rested for two days before downstream assays. All experiments were completed before MSCs reached passage 5.

### 2.2. Cell Microscopy, Intracellular Doxo and Immunophenotyping by Flow Cytometry

Images were taken at 10× magnification using a Nikon Eclipse TS100 inverted microscope (Nikon Instruments Europe B.V., Amsterdam, The Netherlands) or Leica DFC420C digital color camera (Leica Microsystems GmbH, Wetzlar, Germany). BM MSCs were phenotyped upon thawing using CD34, CD45, CD73, HLA-ABC and CD105 monoclonal antibodies (listed in Supplementary Materials Table S1). Doxo content in BM MSCs was measured in the phycoerythrin bandpass filter. Cell cycle analysis of Doxo-exposed BM MSCs was performed using a phase determination kit, according to the manufacturer's instructions (Cayman Chemical, Ann Arbor, MI, USA). Immunophenotyping of differentiated ASCs from CD27$^+$, memory B-cells (MBCs) was assessed using CD20, CD27, CD38 and CD138 monoclonal antibodies (Supplementary Materials Table S1) in addition to cells positive for dead-cell marker (DCM, LIVE/DEAD Fixable cell stain kit Invitrogen). Data was recorded with FACSVerse or LSR II (BD Biosciences, San Jose, CA, USA) and analyzed with FlowJo (TreeStar, Ashland, OR, USA).

### 2.3. Cytokine and Chemokine Array and Protein Network Analysis

Relative levels of 102 cytokine and chemokines in BM MSCs culture supernatants before and after Doxo exposure were assessed with proteome profiler human XL cytokine array kit according to the manufacturer's instructions (R&D Systems, Minneapolis, MN, USA). Relative cytokine and chemokine protein levels on membranes were analyzed in a blinded manner. Captured spot intensity on membranes were graded on a scale between 0 and 10, where 0 represented absence of cytokine, while 10 represented peak cytokine/chemokine level (Supplementary Materials Table S2). Differential spot intensity was estimated as follows: intensities were summed within each donor group (i.e., HC or ALL separately). Proteins corresponding to sums below or equal to the threshold set at the arbitrary level of 5, both before and after Doxo treatment, were discarded. Among the remaining proteins, those having strictly different sums following Doxo exposure were given as input to the STRING tool v11 for analysis and visualization of protein networks.

### 2.4. Quantitative Polymerase Chain Reaction (qPCR)

A QuickRNA MiniPrep kit (Zymo Research, Irvine, CA, USA) was used for total RNA extraction and a SuperScript VILO cDNA Synthesis Kit (Invitrogen) for reverse transcription of RNA. qPCR was performed using Power SYBR Green master mix (Applied Biosystems, Foster City, CA, USA) or KAPA SYBR FAST qPCR Master mix (KAPA

Biosystems Inc., Wilmington, DE, USA). A StepOnePlus Real-Time PCR System (Applied Biosystems) or a Corbett Research RG-6000 Real-time PCR Cycler (Corbett Research, Cambridge, UK) was used to perform and analyze qPCR assays. A minimum of two technical replicates was used. All gene-specific primers were designed in-house (Supplementary Materials Table S3), and the expression values of all genes were normalized to transferrin receptor gene (*TFRC*).

## 2.5. In Vitro Differentiation of ASCs and Co-Culture with BM MSCs

Human adult peripheral blood mononuclear cells from buffy coats were separated by Ficoll-Hypaque gradient centrifugation (GE Healthcare Bio-Sciences AB, Uppsala, Sweden), then processed with an EasySep B-cell enrichment kit, followed by an EasySep CD27$^+$ B cell isolation kit to enrich CD27$^+$ MBCs according to the manufacturer's recommendations (StemCell Technologies, Vancouver, Canada). ASCs were differentiated according to a previously published protocol [28], with minor adjustments of the dose of B-cell activators. Briefly, isolated MBCs cells were cultured in Iscove's modified Dulbecco medium (Invitrogen) containing 50 µg/mL human transferrin, 5 µg/mL human insulin (both from Sigma-Aldrich) and 10% fetal bovine serum (Invitrogen). The cells were seeded at 0.375 × 10$^6$/mL, then exposed to 10 µg/mL human class B CpG oligonucleotide (Invitrogen), 50 ng/mL histidine-tagged soluble recombinant human CD40L and 4 µg/mL anti-poly-histidine mAb (both from R&D Systems), together with recombinant human IL-2 (100 ng/mL), IL-10 (50 ng/mL) and IL-15 (10 ng/mL). At day four, the cells were collected, washed and cultured in a medium containing IL-2 (100 ng/mL), IL-10 (50 ng/mL), IL-15 (10 ng/mL) and IL-6 (50 ng/mL). At day seven, collection and wash of cells was repeated, followed by additional three days culturing in a medium containing IL-15 (10 ng/mL), IL-6 (50 ng/mL) and IFN-$\alpha$ (500 U/mL (all recombinant cytokines purchased from Peprotech, Rocky Hill, NJ, USA). At day 10, cells were collected, and the in vitro 300,000 enriched ASCs were co-cultured with 100,000 Doxo- or non-Doxo-exposed BM MSCs, for four days in total before downstream applications. For IL-6 blocking, either 5 µg/mL of monoclonal antibody (clone 3H3) or isotype-matched control (clone MOPC-21) (Invitrogen and BD Biosciences, respectively) were added to the cultures at day 0. For blocking of GDF-15, 10 µg/mL of human neutralizing GDF-15 antibody (clone 147627) or isotype-matched control (clone 20116, both from R&D Systems) were added at day 0, which was repeated every day until the end of the cell cultures.

## 2.6. ELISA and ELISPOT

The levels of IL-6, IgG (Mabtech, Nacka Strand, Sweden), CXCL12, GDF-15, APRIL and BAFF (R&D Systems) in culture supernatants were measured with ELISA according to each manufacturer's protocols. The detection limits were as follows: 10 pg/mL (IL-6), 0.2 ng/mL (IgG), 30 pg/mL and 47 pg/mL (CXCL12 DuoSet and Quatikine), 7 pg/mL (GDF-15), 30 pg/mL (APRIL), 40 pg/mL (BAFF). Numbers of IgG-secreting ASCs after co-culture were determined by a human IgG ELISPOT kit using MAIPSWU plates (from MabTech and Merc Millipore (Burlington, MA, USA), respectively) in accordance with the manufacturer's instructions. A total of 1000 cells per well were assessed. IgG-secreting ASC numbers were determined following subtraction of spots detected in the negative control wells (PBS coated wells). Plates were analyzed using a CTL-ImmunoSpot S5 Micro analyzer. All conditions were set up in minimum of two replicates.

## 2.7. Statistics

Data were log-transformed (ln) to fit normal distribution before statistical analysis. Paired t-test was used for comparison between the non-Doxo and either of Doxo treatment dosages. *p*-values < 0.05 were considered as statistically significant. All horizontal lines in the dot plots and bars represent median values. GraphPad Prism 8 software (La Jolla, CA, USA) was used for statistical analyses.

## 3. Results

### *3.1. Characteristics of Human BM MSCs Following Exposure to Anthracycline*

As the ex vivo BM MSCs were obtained both from healthy donors (HC) and children who had undergone chemotherapy (ALL), we started by confirming the phenotype of thawed BM-derived MSCs by flow cytometry. Independently of donor status, more than 98% of live CD34$^-$ and CD45$^-$ cells displayed markers compatible with a BM MSC phenotype, as reported previously [27] (Figure 1A, representative phenotype). Doxo exposure at 0.5 μg/mL resulted in preserved MSC morphology, while higher dose exposure (1.0 μg/mL) resulted in lower MSC confluence and swelling (Figure 1B). Taking advantage of the fluorescent properties of Doxo, a positive shift in intracellular Doxo fluorescence intensity could already be detected at a concentration of 0.5 μg/mL, which corresponded to roughly 10% Doxo$^+$ BM MSCs during data acquisition (Figure 1C). Cell cycle analysis showed that exposure to Doxo was paralleled by a lower proportion of BM MSCs in the G1 phase in a dose-dependent manner (Supplementary Figure S1). Altogether, these data indicated that independent of donor status, BM MSCs had a similar phenotype, morphology and sensitivity to the cell-cytotoxic treatment.

To further investigate the effect of Doxo on the function of BM MSC niche by means of its ability to provide soluble survival factors, we performed a screening of multiple cytokines and chemokines in the BM MSCs culture supernatants after exposure to Doxo and used non-treated cells from each donor as baseline controls. The relative amounts of 102 soluble factors were analyzed in six donors ($n = 3$ ALL, $n = 3$ HC) by a semiquantitative approach (Supplementary Table S2), showing significantly diminished CXCL12, but increased IL-6 and GDF-15 following Doxo exposure (Figure 1D, representative protein arrays), irrespective of stratification into ALL and HC (Supplementary Table S2). To further comprehend the dynamics proteins released from HC and ALL donor BM MSCs after Doxo exposure, the protein array data were used to create a protein-association network for upregulated and downregulated proteins (Figure 1E). Despite HCs displaying a more diverse set of cytokines and chemokines affected by Doxo compared to ALL patients, in both investigated groups altered levels of CXCL12, IL-6 and GDF-15 appeared as primary PC niche factors for further evaluation. Of note, we did not detect any substantial levels of BM MSC derived APRIL and BAFF in HC or ALL donors prior or after in vitro Doxo exposure (Supplementary Figure S2).

The relative amounts of cellular CXCL12, IL-6 and GDF-15 mRNA and proteins in BM MSCs cultures were examined in HC and ALL donors. CXCL12 mRNA levels were significantly decreased following 0.5 and 1.0 μg/mL Doxo exposure (Figure 2A, $p < 0.001$ and $p < 0.01$, respectively) which paralleled with decreased protein levels (Figure 2B, $p < 0.01$ for both 0.5 and 1.0 μg/mL Doxo). Conversely, Doxo exposure increased the expression of IL-6 both at the level of gene expression (Figure 2A, $p < 0.05$ and $p < 0.01$ for 0.5 and 1.0 μg/mL drug dose, respectively) and protein expression (Figure 2B, $p < 0.01$ for 0.5 μg/mL Doxo). Similar to IL-6, there was a dose-dependent increase of GDF-15 mRNA and protein (Figure 2A,B, $p < 0.001$ for 0.5 and 1 μg/mL Doxo). The results did not differ between the HC and ALL subjects, and we could not detect any bias in CXCL12, IL-6 or GDF-15 expression profiles related to previous chemotherapy in the latter group.

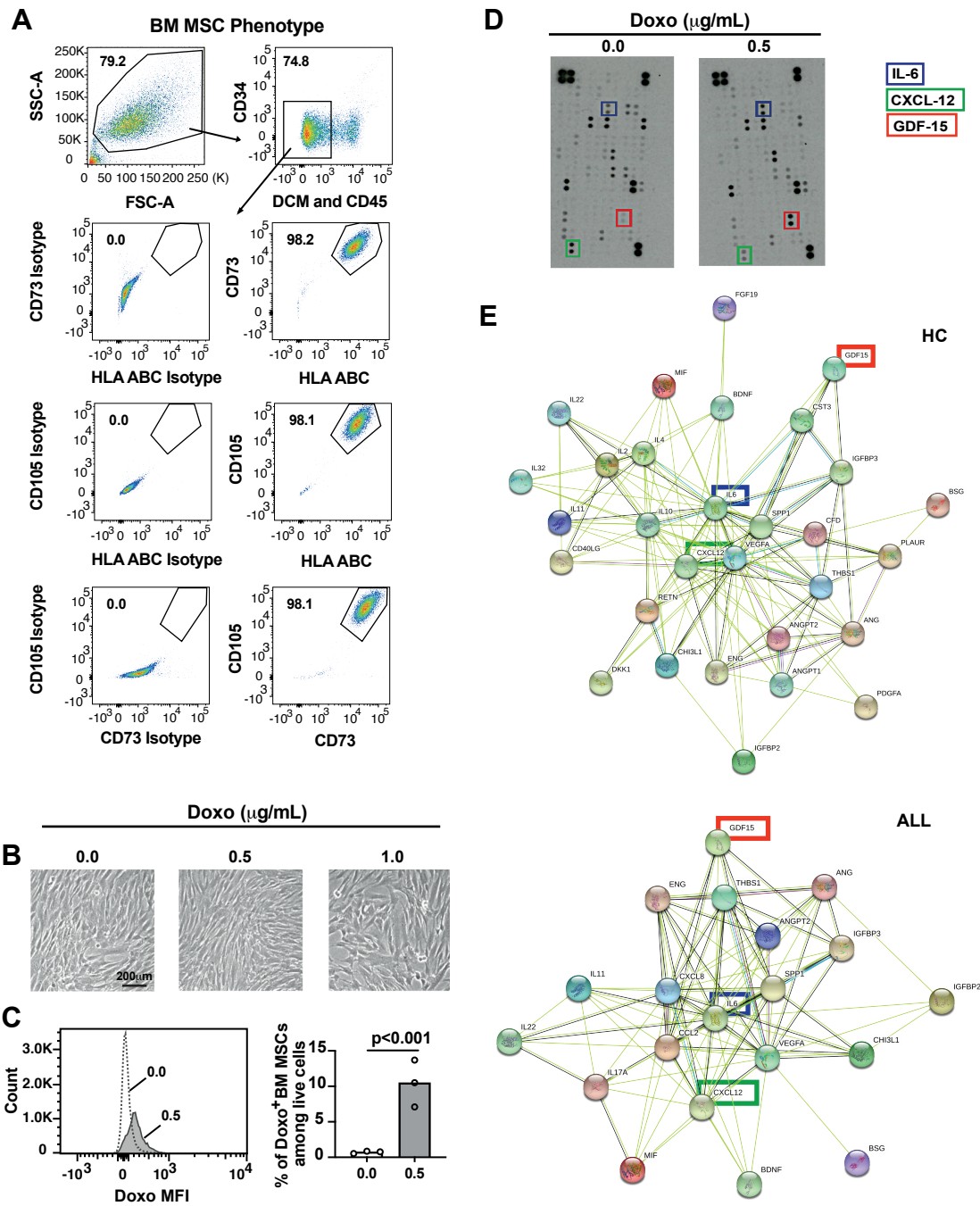

**Figure 1.** Primary human bone marrow (BM) mesenchymal stromal cell (MSC) characteristics following in vitro exposure to doxorubicin. (**A**) Representative gating-strategy of live (DCM⁻) single BM MSCs defined as CD45⁻ and CD34⁻ and CD73⁺, CD105⁺ and HLA-ABC⁺ cells ahead of doxorubicin (Doxo) exposure (ALL donor). (**B**) Microscopy images of BM MSCs exposed to 0.0 µg/mL, 0.5 µg/mL or 1.0 µg/mL Doxo, at 10× magnification (Nikon Eclipse TS100 inverted). (**C**) Representative histogram on Doxo fluorescence intensity per Doxo⁺ cell and compiled data of % Doxo⁺ BM MSCs measured by flow cytometry (*n* = 3, HC donors). (**D**) Representative picture of membranes displaying BM MSC expression of an array of cytokines and chemokines before (0.0 µg/mL) and after (0.5 µg/mL) Doxo exposure (*n* = 3 HC and *n* = 3 ALL donors). Data from proteins that were differentially expressed after Doxo exposure were used for visualization of (**E**) protein–protein networks that were generated for each group (HC and ALL) separately. Nodes of the network are proteins indicated by their gene symbol, and edges are known protein–protein associations. Blue/purple edges are known interactions, while green/red edges are predicted interactions. Node colors are random and serve no particular meaning.

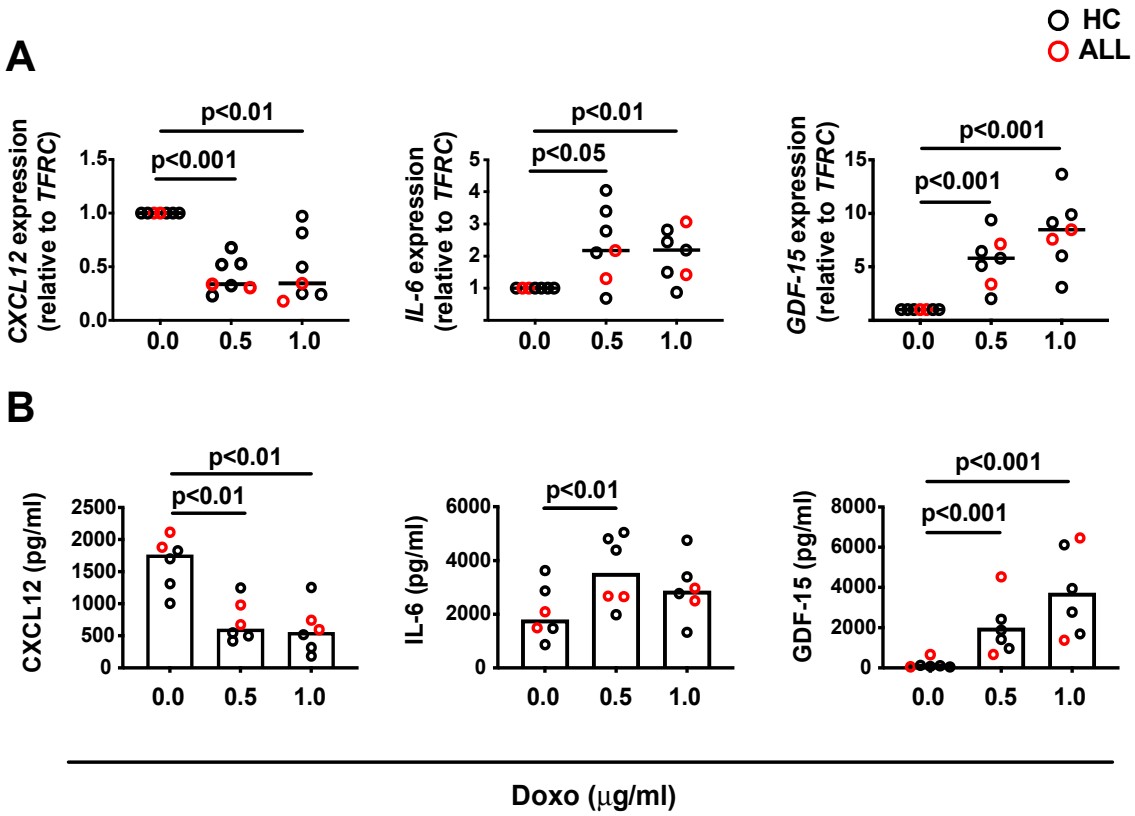

**Figure 2.** Alterations in CXCL12, interleukin 6 IL-6 and growth and differentiation factor 15 (GDF-15) gene and protein expression by BM MSC following in vitro exposure to doxorubicin. (**A**) Relative *CXCL12*, *IL-6* and *GDF-15*, mRNA levels normalized to *TFRC*, quantified by real time PCR in BM MSCs following Doxo exposure (*n* = 7 donors). (**B**) CXCL12, IL-6 and GDF-15 protein levels in the BM MSC supernatants after Doxo exposure, measured by ELISA (*n* = 6 donors). (**A**,**B**) Data acquired during two independent experiments. **O**: Healthy controls, **O**: Children under treatment for ALL

### 3.2. Doxo-Exposed Human BM MSCs Retain the Capacity to Support IgG-ASCs In Vitro

Since Doxo influenced the ability of BM MSCs to produce known PC survival factors, next we evaluated how Doxo-exposed BM MSCs affected the persistence of ASCs. To get access to an enriched pool of ASCs, healthy blood donor MBCs were used for differentiation of ASCs using an established in vitro protocol (Jourdan 2009) (Figure 3A, schematic overview of setup). By day 10 of ASC differentiation, roughly 75% of living cells had gained marked CD38 expression showing a plasmablast phenotype, while approximately 30% displayed a phenotype compatible with terminally differentiated LLPCs (CD20⁻CD38⁺CD138⁺, Figure 3B). The enriched ASCs were co-cultured with previously Doxo-exposed or non-exposed BM MSCs (*n* = 4 donors), after which IgG-producing ASCs were evaluated (Figure 3C). Surprisingly, we found higher proportions of IgG⁺ ASCs in co-cultures where BM MSCs had been exposed to Doxo compared with donor matched samples in medium only (Figure 3C), which was paralleled with a similar pattern of soluble IgG concentration in corresponding ASC and BM MSC co-culture supernatants (Figure 3C). Compared to the isotype control, neutralization of IL-6 resulted in a significant, yet partial, reduction in the number of IgG-producing ASCs (Figure 3C, *p* < 0.05 for both 0.5 and 1.0 µg/mL), and minor decrease in soluble IgG in ASC and Doxo-exposed BM MSC co-culture supernatants (Figure 3D, *p* < 0.01 for 1.0 µg/mL Doxo). Neutralization of GDF-15 alone, or in combination with IL-6, did not influence IgG-producing ASC proportions or IgG levels in co-cultures (data not shown).

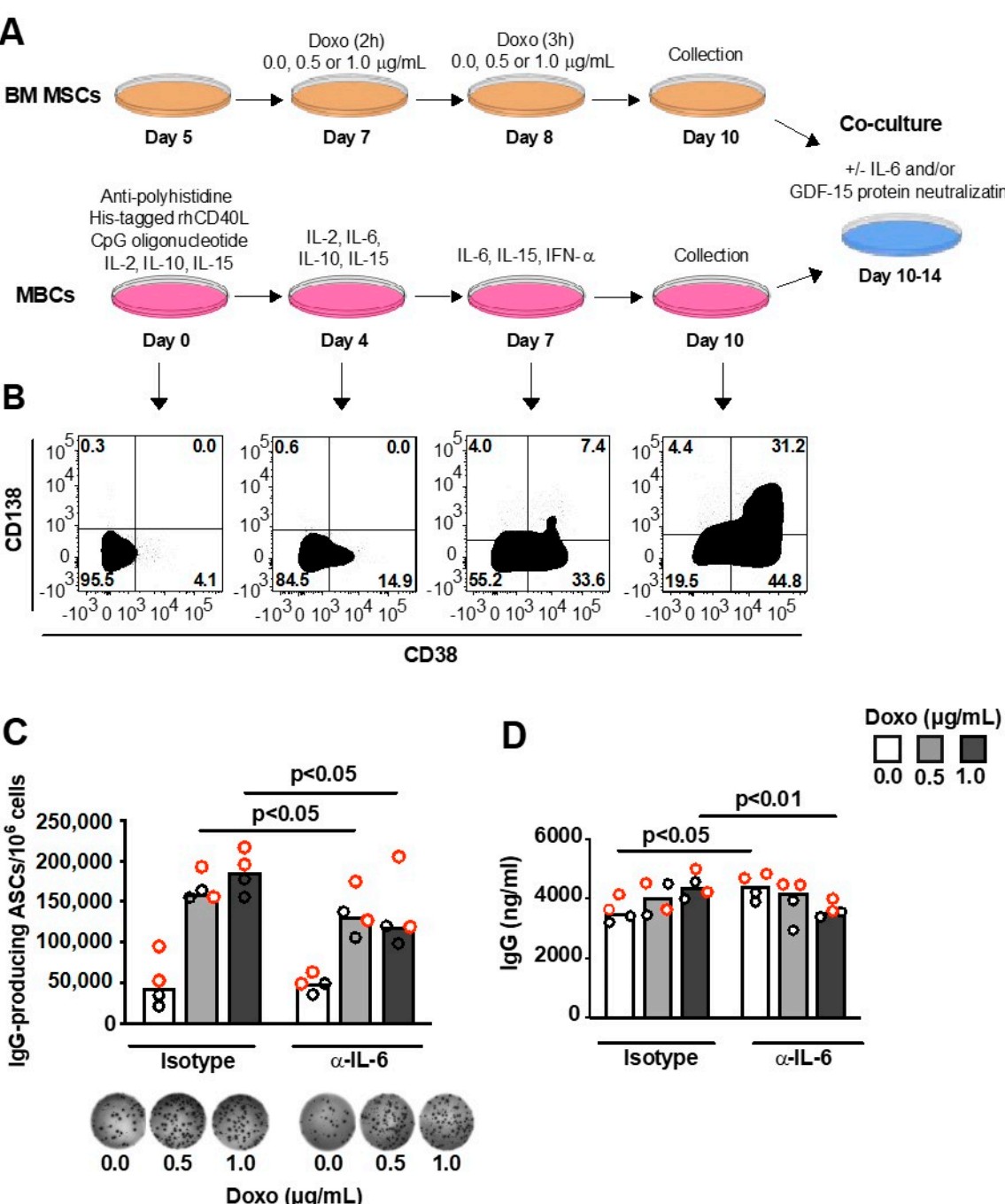

**Figure 3.** Doxo-exposed primary human BM MSCs sustain a higher number of IgG-producing antibody-secreting cells (ASCs). (**A**) Schematic picture of ASC enrichment following differentiation of memory B-cells (MBCs), and their co-culture with BM MSCs. At day 10, the enriched ASC population (pooled from *n* = 2 donors) was collected and co-cultured with earlier Doxo-exposed or non-exposed BM MSCs (*n* = 2 HC and *n* = 2 ALL) for additional four days (day 10–14). (**B**) Representative phenotype of ASCs (CD38$^+$CD138$^-$ and CD38$^+$CD138$^+$) which gradually became enriched in the in vitro culture from day 0 to 4, 7 and 10 of differentiation. (**C**) Representative wells and compiled data of IgG spot-forming ASCs evaluated by ELISPOT after BM MSC and ASC co-culture with or without IL-6 neutralization. (**D**) Soluble IgG levels in BM MSC and ASC co-cultures, with or without IL-6 neutralization. **O**: Healthy controls, **O**: Children under treatment for ALL

## 4. Discussion

An extensive amount of research has previously focused on the ASC survival niche within the BM [25], which is central for the maintenance and protection of serological memory. However, even if there is increasing knowledge about BM MSCs with a focus

on their gene and protein expression profiles [29,30], the understanding of how the BM niche influences memory cell preservation during steady-state or during toxic conditions is still far from complete. In mice, age has been connected to the capacity of BM to provide proper support for plasmablast homing and/or conversion into LLPCs [31]. Similar studies in pediatric populations are not feasible, as sampling of BM is performed mainly for the purpose of disease monitoring, which hampers the understanding of the PC niche in otherwise healthy humans at a young age. We assessed the phenotype and growth characteristics of the ex vivo BM-derived MSCs from adult HC and pediatric ALL donors and found no differences between the two groups. Also, in humans, pediatric and adult MSCs differ by means of proliferation and functional characteristics even if findings from different studies are inconsistent by means of cell source and functional readouts. Results herein are in part supported by earlier findings showing that children <5 years of age who underwent hematopoietic stem cell transplantation reconstituted their BM-derived MSCs to normal levels [32]. On the other hand, another study showed that mesenchymal stem cells isolated from adults (20–50 years old) and pediatric (2–13 years old) donors displayed no morphological differences or capacity to modulate antigen presentation, while mesenchymal stem cell growth capacity was strictly related to the age of donors [33]. Following assessment of BM MSCs from 13–80 years old individuals, those derived from younger donors showed, among others, increased MCAM, VCAM-1, PDL-1 and CD71, and lower IL-6 production when co-cultured with activated T cells [34].

A consensus exists that Doxo, like most anthracyclines, is a nonselective drug inducing apoptosis in rapidly proliferating cells [35]. Since BM MSCs exhibit a low cellular turnover rate *in vivo*, it has been suggested that they are resistant to cell-cytotoxic treatment [36]. It should, however, be emphasized that BM-derived MSCs from ALL donors were taken half a year after the initiation of cancer treatment, which does not exclude the possibility that the MSCs would be more negatively affected by ALL treatment at cessation of therapy, which goes on during approximately 2.5 years. This is in keeping with that the type, dose and duration of chemotherapy regimen play a significant role in damage to the BM compartment following cancer treatment [37,38]. We registered a marked fraction of live Doxo$^+$ BM MSCs two days after drug exposure. This indicated that the rate of Doxo metabolism and efflux by human BM MSCs may be a slower process compared with results from animal and tissue models showing a cellular turnover rate of 10–30 min [39,40]. In vitro studies of Doxo-exposed human BM MSCs show that apoptosis manifests through excessive generation of reactive oxygen species [41] or direct interference to DNA double helix [42] leading to cell cycle arrest. Upon Doxo exposure, we found a significantly lower percentage of BM MSCs at G1 phase, opposite to an increased pattern of cells arrested in G2 phase, which is in accord with interruption of DNA synthesis [42]. At this stage, however, the long-term fate of MSCs in the G2 phase remains unclear. Furthermore, our findings are contrasted by those made by Kozhukharova et al. who reported that Doxo treatment decreased the number of mesenchymal stem cells in the S-phase while increasing numbers of cells arrested in the G0/G1 phase [43]. The reason for the discrepancies between findings in this study and that in ours are unclear, but could in part relate to differences in drug dose and exposure time,

To understand how cytokine and chemokine production by BM MSCs from ALL and HC donors were affected by Doxo exposure, and if the recorded changes could be implicated in ASC longevity, we screened for multiple cytokines and chemokines, and found that IL-6, CXCL12 and GDF-15 were strongly influenced by Doxo treatment. A subsequent protein network analysis—allowing us to assess and visualize a relatively large data set by an unbiased manner before identification of soluble factors that have been acknowledged as central for PC survival [7,8,12]—confirmed these findings. Overall, the HC group showed a more diverse protein network in response to Doxo exposure than the ALL group. A cytokine that was differentially expressed only in the ALL group was IL-17A, which has been implicated in maintenance of ASCs in inflammatory tissues of mice [44]. Thus, we doubt the implication of IL-17A in IgG responses and human BM

PC niche. A diminished CXCL12 in response to chemotherapy might be unfavorable for vaccine-induced memory in patients with ongoing cancer treatment. Nonetheless, increased cancer cell infiltration of BM niche due to excessive CXCL12-CXCR4 signaling, and thereby potential out-competing of the normal protective LLPCs in favor of malignant cells, should be taken into consideration [25].

Surprisingly, we found elevated proportions of IgG ASCs upon Doxo-induced decrease of CXCL12 by BM MSCs, suggesting that additional or alternative components might play a part in long-term PC survival under compromised conditions like chemotherapy. IL-6 is a multifaceted cytokine that is, for example, produced upon cellular stress through activation of the NF-kβ pathway [45], but also mandatory for the PC niche and in vitro generation and survival of human LLPCs [13]. Since elevation of IL-6 and IgG-ASC numbers coincided, while CXCL12 levels were diminished after Doxo exposure, we reasoned that IL-6 might compensate the role of CXCL12 in promoting ASCs sustenance. Neutralization of IL-6 was connected to a marked decrease in IgG-ASC numbers in the co-cultures with Doxo-exposed BM MSCs. In our hands, the net balance between depressed CXCL12 and enhanced IL-6 in BM MSC and ASC co-cultures may be one explanation behind the sustained numbers of ASCs [13]. We also observed that the cytokine GDF-15 was remarkably enhanced in response to Doxo, which could further contribute to the higher numbers of ASCs. Since GDF-15 was previously reported to promote the growth, survival and self-renewal of malignant PCs in myeloma patients [19,20], we asked if GDF-15 contributed to the maintenance of normal ASCs. Our in vitro results were not in favor of a marked role for GDF-15 in ASC survival. This finding should, however, be interpreted with caution. Unlike IL-6, neutralization of GDF-15 was technically challenging. This is due to the limitation in available and well-performing reagents on the market. Furthermore, a recent study proposed that human BM MSC-derived fibronectin was an alternative factor essential for ASC longevity and homing to BM [29]. Even if the relationship between IL-6 and fibronectin is unclear in the BM niche, IL-6 presence has been connected to the reduction of fibronectin in hepatocytes [46]. If we extrapolate this to our in vitro setting, then fibronectin's role in sustaining ASCs becomes questionable. Likely, the requirements for survival of ASCs during their differentiation into LLPCs may be different [47], and the relative importance of each subset of MSCs, such as the CXCL12$^+$ reticular stromal cells [6,48], for the persistence of ASCs deserves further attention.

We acknowledge that a relatively short culture follow-up time as well as use of one single component in ALL treatment protocol was a shortcoming that influenced our findings. ASCs can survive up to seven days when cultured together with BM MSC-obtained secretome that is deprived of select factors know to promote PCs [29]. Thus, we cannot rule out the possibility that IL-6 and/or GDF-15 mediated effects on ASC longevity and antibody release would have been more pronounced with increasing co-culture time. Nevertheless, our study is unique in that it investigates the capacity of PC survival niche to support normal ASC under the influence of chemotherapy exposure using both otherwise healthy BM MSCs and those derived from children under ALL therapy. While deprivation of IL-6 was connected to partial reduction of IgG-producing ASCs, diminished CXCL12 and GDF-15 were not enough to alter ASC numbers. This suggests that the PC survival niche uses redundant mechanisms to sustain the mediators of serological memory under BM suppression, and that serological memory loss may arise during late stages of ALL therapy.

**Supplementary Materials:** The following are available online at https://www.mdpi.com/2673-6357/2/1/9/s1. Table S1. List of monoclonal antibodies used for flow cytometry. Table S2. Soluble proteins derived from in vitro cultures of Doxo-exposed and non-exposed human bone-marrow derived mesenchymal stromal cells. Table S3. Oligonucleotides used as qPCR primers. Figure S1 Cell cycle analysis following in vitro exposure of bone-marrow mesenchymal stromal cells to Doxorubicin. Figure S2. Bone-marrow mesenchymal stromal cells produce minor levels of APRIL and BAFF regardless of Doxorubicin exposure.

**Author Contributions:** Conceptualization, A.N. and S.S.-H.; methodology, S.S.-H., G.L., A.H., D.U., M.S. and A.Y.; software, S.S.-H., G.L. and R.C.; validation, A.N., S.S.-H., G.L. and E.S-E.; formal analysis, A.N., S.S.-H., G.L., D.U., D.S., R.C. and E.S-E.; investigation, A.N. and S.S.-H.; resources, A.N., M.S. and S.S.-H.; writing-original draft preparation, A.N., S.S.-H., G.L., M.S., A.H. and E.S.-E.; writing—review and editing, A.N., S.S.-H., G.L., A.H. and E.S.-E.; supervision, A.N. and S.S.-H. All authors have read and agreed to the published version of the manuscript.

**Funding:** The study was supported by grants from the Swedish Research Council (2017-02001), Childhood Cancer Foundation (2014-0112), Märta and Gunnar V. Philipssons stiftelse, the Stockholm County through the ALF agreement (2015-0239), Karolinska Institute Foundations & Funds, Cancer Foundation (CAN 2017/460) and Cancer and Allergy Foundation (204).

**Institutional Review Board Statement:** The study was conducted according to the guidelines of the Declaration of Helsinki, and by the Regional Ethics Review Board in Stockholm (2016/458-31).

**Informed Consent Statement:** Informed consent was obtained from all subjects involved in the study.

**Data Availability Statement:** Not applicable.

**Conflicts of Interest:** The authors have no conflicts of interests.

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
