# Peer review of "Human Bone Marrow Mesenchymal Stromal Cell-Derived CXCL12, IL-6 and GDF-15 and Their Capacity to Support IgG-Secreting Cells in Culture Are Divergently Affected by Doxorubicin"

_hemato, doi:10.3390/hemato2010009_

Round 1
Reviewer 1 Report
In this study, the authors asked whether cell cytotoxic treatment alters the capacity of human bone marrow mesenchymal stromal cells (BM-MSCs) to support the survival of antibody-secreting cells (ASCs). They profiled a large number of cytokines and chemokines from BM-MSC derived from healthy donors and treated ALL children following exposure to Doxorubicin. They also investigated the impact of Doxo-induced modifications on the MSC ability to support the Ig-producing ASCs. They observed alterations in IL-6, CXCL12 and GDF-15 after DOXO treatment but these alterations are not sufficient to disintegrate IgG-ASC support. They conclude that serological memory loss may arise during late stages of ALL therapy.
Overall the paper is interesting and the conclusions arrived at in this paper are, I think, justified. The paper is well written and results are clearly presented. The paper appears of interest for the journal’s readership. However the manuscript needs several minor and major revisions.
The major criticism of this study concerns the age of healthy donors. MSC were isolated from healthy ADULT donors (n=3) and children (n=3). The age of healthy donors must be indicated but the comparison between adult versus pediatric MSCs does not seem optimal. In the discussion, the authors say that similar studies in paediatric populations are not feasible. However in the case of autologous hematopoietic stem cell transplantation, it is possible to use the washouts of discarded collection sets, left over at the end of the filtration of bone marrow explants derived from young donors (Capelli C et al, Cytotherapy 2009).
In general, in comparison with adult MSCs, pediatrics MSCs display higher proliferation, better differentiation abilities and the functional factors they secrete may also change. The authors should comment this crucial point.
MSCs were treated 2hours with DOXO and after washed. After overnight rest in fresh medium, MSCs were treated 3hours with DOXO. After wash and 2 day incubation in fresh medium, MSCs were evaluated. The authors should explain the design of DOXO treatment of MSCs.
Proteins having strictly different sums following DOXO exposure were analyzed for protein networks (STRING tool). However the representative networks seen in figure 1E are unreadable.
The in vitro enriched ASCs were co-cultured with BM-MSCs for 4 days. What is the cell-cell ratio evaluated in these co-cultures?
Higher DOXO dose exposure (1µg/ml) resulted in lower MSC confluence and swelling. What is the viability of these treated MSCs? Low DOXO dose causes MSC premature senescence. Did you observe senescent cells in your experiments?
Upon Doxo exposure, the authors observed a significant lower percentage of MSCs at G1 phase and an increased pattern of cells arrested in G2 phase. However, in the study of Kozhukharova I et al (Int J of Hematology 2018), after DOXO treatment (0.1µM), the number of cells in S-phase drastically decreased and the cells mostly accumulated in G0/G1 phase. The authors should comment these discrepancies.
Author Response
Kindly refer to attachment.

Reviewer 2 Report
In the current article, Lasaviciute and colleagues investigate the effects of a commonly used chemotherapy agent, doxorubicin, on human bone marrow-derived mesenchymal stromal cells (MSCs). In particular, the authors investigated the effects of doxorubicin on MSC cytokine production, which in turn may affect the survival of antibody secreting cells (ASCs). In a wider perspective, this question may help to clarify the mechanism by which pediatric patients treated with chemotherapy have an impaired vaccine-induced antibody production, a question of particular interest given the global pandemic and the mass vaccination effort currently underway.
In spite of these interesting premises, the current study only offers a partial answer to the effects of chemotherapy on the interplay between the BM-MSC niche and ASCs. One of the crucial limitations of the study is that it investigates only the short-term effects of doxorubicin exposure and it suggests that the alterations observed in the levels of two important cytokines are not sufficient to disrupt the niche support to ASCs. The narrow window of observation and the limited number of molecules assessed is, in my opinion, a major impairment to the relevance of the study.
Here are some suggestions to improve the quality of the paper
- the authors should better specify the features of the clinical samples used. Since the ALL samples were collected at day 106 of treatment, how good was the response at that point? Please specify the disease burden at the time of collection.
- The resolution of figure 1 is so poor that it is barely readable
- In order to obtain a more comprehensive analysis of the short-term effects of doxorubicin treatment, the authors should consider profiling the transcriptome of Ig-producing ASCs co-cultured on of MSCs (with or without exposure to doxo treatment).
- An in vivo modeling of the proposed observations would greatly increase the interest of the study. Treatment of experimental animals with doxorubicin could be used to assess the short and long-term effect of chemotherapy on niche populations as well as on ASCs, which in turn will allow for a better characterization of the immune response.
Reviewer 3 Report
In this study, the authors demonstrated that bone marrow-derived mesenchymal stem cells (BM MSCs) by co-cultured with Doxorubicin (Doxo) were not sufficient to disintegrate the support of IgG-antibody secreting cells (ASCs). This is interesting study for the therapeutic aspects of chemotherapy (ALL). The experimental design is appropriate to prove authors’ hypothesis and the results are solid. Discussion section also deals with appropriate interpretation of the present and related previous studies. Therefore, this manuscript might be appropriate for its publication after satisfying following minor concerns.
- Since the title is too specific, it needs to be modified to a general title that can include the content.
- The quality of Figure 1 is too low that contents cannot be interpreted.
- Clarify the definition of ‘ALL’, especially in the abstract.
- The dilution rate is not included in the antibody information (Method section and Table 1).
- In Figure 1B, scale bar was not shown.
- In Figure 1D, the authors displayed the expression of an array of cytokines and chemokine, however the information on proteins other than IL-6, CXCL-12, and GDF-16 was not displayed. Also, the author should be analyze and discuss the data through graphs for relative comparison of IL-6, CXCL-12, and GDF-16.
- For understanding BM niche influences memory cell preservation during steady-state or during toxic conditions, the author should be examine the confirmation of intracellular level.
Round 2
Reviewer 1 Report
The authors have adressed most comments raised during the review in an adequate fashion.
Reviewer 2 Report
After this round of revisions, the authors significantly improved the graphical quality and methods of the presented data. I appreciated the explanation of the technical difficulties that hampered the collection of a wider set of data, also considering that the work is done on primary samples.
I believe that this paper is a fair starting observation to justify further studies on stromal effect on antibody secreting cells. These additional studies would ideally comprise in vivo modeling and niche dissection by single-cell molecular profiling, as well as in vivo microscopy. These experiments may go beyond what requested by the current publication, but I encourage the authors to further expand their observations in such directions.